# The Relationship between the Strength Characteristics of Cerebral Aneurysm Walls with Their Status and Laser-Induced Fluorescence Data

**DOI:** 10.3390/biomedicines9050537

**Published:** 2021-05-12

**Authors:** Elena Tsibulskaya, Anna Lipovka, Alexandr Chupakhin, Andrey Dubovoy, Daniil Parshin, Nikolay Maslov

**Affiliations:** 1Khristianovich Institute of Theoretical and Applied Mechanics SB RAS, 630090 Novosibrsk, Russia; euglenaria@gmail.com (E.T.); nmaslov@itam.nsc.ru (N.M.); 2Lavrentyev Institute of Hydrodynamics SB RAS, 630090 Novosibirsk, Russia; alexander190513@gmail.com (A.C.); dubovoy@mail.ru (A.D.); danilo.skiman@gmail.com (D.P.); 3Federal Neurosurgical Center, 630048 Novosibirsk, Russia

**Keywords:** cerebral aneurysm, laser-induced fluorescence, tensile test

## Abstract

Background: Cerebral aneurysms (CA) are a widespread vascular disease affecting 50 per 1000 population. The study of the influence of histological, morphological and hemodynamic factors on the status of the aneurysm has been the subject of many works. However, an accurate and generally accepted relationship has not yet been identified. Methods: In our work, the results of mechanical and spectroscopic measurements are considered. Total investigated 14 patients and 36 their samples of CA tissue. Results: The excitation–emission matrix of each specimen was evaluated, after which the strength characteristics of the samples were investigated. Conclusions: It has been shown that there is a statistically significant difference in the size of the peaks of two components, which characterizes the status of the aneurysms. In addition, a linear regression model has been built that describes the correlation of the magnitude of the ultimate strain and stress with the magnitude of the peaks of one of the components. The results of this study will serve as a basis for the non-invasive determination of the strength characteristics of the cerebral tissue aneurysms and determination of their status.

## 1. Introduction

An intracranial or cerebral aneurysm (CA) is a dangerous pathology of the brain vascular system and is diagnosed for 2–5% of the population [1]. The sudden rupture and delay in its treatment lead to disability or death. However, the risks of aneurysm rupture and postoperative complications are comparable. This makes the decision for aneurysms treatment controversial since it is unclear whether the aneurysm is going to rupture. The assessment of CA rupture risk is one of the biggest challenges in modern neurosurgery.

The study of the mechanical properties of vascular tissues is necessary for making accurate prognosis. It is a complicated task due to the complex layer structure of a vessel, where each tunica has unique protein composition, which varies during pathology development. Thus, having a group of integral parameters that describe tissue strength characteristics could be an aid in preoperative modeling of cerebral hemodynamics. Its results should allow to perform surgery in the most effective way or avoid it if possible. Such parameters could be determined using different methods, such as strength tests [2], histological studies [3]. In the meantime, nondestructive methods are preferable.

Intrinsic laser-induced fluorescence (LIF) is already known to be of use in atherosclerosis and valve calcinosis [4] diagnostics and the control of decellularization biotechnological process [5]. The main advantage of this method is that it does not require extra fluorescent labeling. Vessels contain several fluorophores, such as amino acids and collagen cross-links, which are connected to the mechanical properties of tissue. LIF spectra obtained using ultraviolet laser radiation below the 300-nm region are still not well studied and can give data on the state of the vessel tissue.

In this paper, we performed an experimental study of the artery walls with cerebral aneurysms using tunable in 210–355 nm laser radiation. Two types of tissues were used: ruptured and unruptured cerebral aneurysms. The measured excitation–emission matrices were analyzed using the narrow peak modification of principal component analysis.

The relationships of the energy peaks of fluorescent components with the status of the aneurysm, as well as with the strength characteristics, were investigated. Work on the study of the strength properties of half of the samples was presented in [6].

## 2. Materials and Methods

The fluorescence of the samples under study was induced by a pulsed ultraviolet laser with optical parametric oscillator “Vibrant HE 355 II + UV” (Opotek, Carlsbad, CA, USA), tunable in the wavelength range of 210–355 nm. A 5% beam-splitter was used to divert a part of the laser beam to a photodiode for laser pulse energy measurement. The laser beam was pointed toward the tissue sample located in a stainless-steel cell. The stainless-steel cell shows almost no fluorescence. The sample fluorescence was directed by a spherical mirror onto the entrance slit of the spectrometer “Action SP2300” (Princeton Instruments, Trenton, NJ, USA) with cooled CCD matrix with open electrodes “Pixis 256” (Princeton Instruments, Trenton, NJ, USA). The entrance slit of the spectrometer was at the focus of the mirror. Thus, the fluorescence spectrum was measured averaged over the surface of the sample. Excitation-emission matrices were scanned using laser radiation in the range of 210–350 nm with a step of 10 nm. To block the laser radiation and to avoid CCD damage, an optical filter was placed in front of the spectrometer’s entrance slit. For the 210–290 nm tuning range, a custom polymer high-pass filter with a 300 nm border was used. For the 300–310 nm range, an acetone high-pass liquid filter was used, providing a 330-nm cutoff wavelength. For the 320–330 nm range, a high-pass filter made from BS8 glass was used, with a 400-nm cutoff wavelength. Spectrum measurements were performed 5 times for each excitation wavelength. Then each spectrum was normalized to the total absorbed laser energy, averaged and normalized to the spectral sensitivity of the device, measured by deuterium and tungsten lamps. The laser pulse energy was limited to the level of 200 µJ/cm^2^. The fluorescence intensity nonlinearly depends on the laser pulse energy [5]. If this energy is exceeded, it may introduce a substantial data discrepancy during the normalization of spectra on the laser pulse energy. If the laser pulse energy is limited, the standard deviation of the normalized fluorescence intensity did not exceed 5%. The total irradiation dose was in the 1–10 mJ/cm^2^ range for each excitation wavelength. It was also shown [7] that damage caused in such conditions by tunable laser radiation does not significantly influence measured excitation–emission matrices. More details about the measurement system can be found in [8].

The samples of cerebral aneurism tissues were resected during microsurgical clipping at the Federal Neurosurgery Centre of Novosibirsk. The tissues were preserved in 0.9% sodium solution at +2–+4 °C while being transported [9]. Mechanical tests were conducted on two universal tensile machines: Zwick and Roell Z10 (Ulm, Germany) and Instron 5944 (Norwood, MA, USA). The Instron machine is equipped with a biobath that allows the specimen to be tested under the conditions that most resemble the natural environment. The biobath is filled with sodium chloride solution prior to the experiment and heated up to 37 °C. The sample is immersed in the solution during the entire loading process. While being tested on the Zwick and Roell machine (Ulm, Germany), the specimen was thoroughly moisturized before and during the loading by being sprayed with a sodium solution.

After the delivery to the laboratory, the geometry of the specimen is measured. Then it is placed in specially made clamps, wrapped with waterproof sandpaper, that prevents the slipping of the specimen. The clamps then are placed in the jaws of the tensile machine.

The loading was conducted with a constant pulling speed of 1 mm/min on the Zwick and Roell machine (Ulm, Germany) and 2 mm/min on the Instron machine. Each machine was programmed to perform several stages of loading. The displacement on the first stage was 0.25 mm and was increased with each stage by 0.25 mm. During the first five stages, preconditioning [10] was used, which allows correctly measuring the stresses in the tissue. This experimental approach to testing is widely used with arterial tissues, including intracranial aneurysms [11,12,13].

Each sample characteristic size was about 5–6 mm. Twenty samples were studied; the excitation–emission matrix of each specimen was twice measured on the intima’s side. Between the measurements, the specimen was rotated to average fluctuations caused by variability of the radiation collection geometry.

Statistical analysis was performed in a Python environment using the following libraries: numPy, pandas, matplotlib, seaborn, patsy, and sklearn.

## 3. Results

A typical excitation–emission matrix of brain vessels with aneurysms is presented in Figure 1 in the form of a waterfall plot. It is seen that all spectra are continuous with broad bands. The spectral shapes and overall intensities depend on the excitation wavelength. This means that several fluorophores contribute to emission spectra. The fluorophores’ bands heavily intersect.

For 210–290 nm excitation, all spectra show the main peak around 325 nm, which could be attributed to tryptophan fluorescence. However, the peak maximum can shift in the 320–330 nm spectral region, depending on the sample and excitation wavelength. This means that at least two different fluorophores take part in this peak formation. It could be two forms of proteins containing tryptophan—the spectrum of the latter depends on its surroundings. Tyrosine also could be a cause of such spectral alterations. In the longer wavelength region, all spectra are different but without distinct features.

For 300 and 310 nm excitation, due to the filter used, parts of the measured spectra with wavelengths shorter than 340 nm are deformed. In the longer wavelength region, the peak with 374–380 nm maximum could be observed. The 374–380 nm peak appears to be due to crosslinks in collagen and elastin, and its features also could depend on the protein structure [14]. Once again, the true shape of this fluorophore spectrum is unclear because, for these excitation wavelengths, its spectral band intersects with the tryptophan band from the short wavelength side and various fluorophores from the longer wavelength side.

For 320 and 330 nm excitation, the high pass filter deforms spectra for wavelengths shorter than 400 nm. Thus, collagen fluoresce could not be fully observed—it is a tradeoff for protecting the spectrometer’s sensor in the experiments with changing wavelengths. The longer wavelength tail of collagen fluorescence forms a peak with a 400 nm maximum. Its longer wavelength side differs from sample to sample, sometimes forming local maxima around 450 and 550 nm. It could also appear due to the crosslinks in collagen [15,16].

The complex excitation–emission matrix structure does not allow performing any analysis without regression to individual fluorophores’ spectra. The above-mentioned variability of possible fluorophores contributing to overall spectra makes separate measuring the correct spectra of individual components an extremely complicated task. Thus, each fluorophore spectrum could be isolated only using measured spectra of tissues.

There are various methods of analyzing large sets of data without a previously known internal structure. One of the most popular ones is the principal component (PC) analysis. It allows one to reduce the dimension of the data array to a statistically meaningful array of scores with a significantly lower number of items. Here it means finding spectra of individual components using data diversity. For the 210–290 nm excitation range, Figure 2 shows six PCs, which describe most of the normalized measured spectra with RMS deviation less than 0.01. These components are enough to find the differences between spectra, but this method has a substantial disadvantage. PCs are abstract functions, alternating by construction. They do not directly correspond to real fluorophores, being linear combinations of those.

Finding the fluorophores’ spectra themselves is a separate task. Since any positively defined linear combination of principal components can claim the role of the fluorophore spectrum, additional considerations are necessary for their choice. If the sample is optically thin, one can take advantage of each excitation–emission matrix being a correlated set of data. In most cases, due to vibrational relaxation, the spectrum shape of each fluorophore does not depend on the excitation wavelength. For an optically thin sample, this means that contribution of each fluorophore is proportional to excitation and emission spectra product, making various methods of tensor PC analysis applicable [17]. However, in this case, all attempts to apply such methods failed, meaning each spectrum should be analyzed separately. Various multivariate curve resolution (MCR) methods allow finding positively defined individual components [18]. However, these components match the spectra of pure fluorophores only if the latter meet Manne’s conditions [19]. This is not true in our case; the components heavily intersect. It may lead to some uncertainty in the reconstructed spectra; the results of various iterative methods, such as alternating least squares (MCR-ALS), depend on the initial estimates. The narrow peak approach [20] allows finding unique representation. It is based on the consideration that the continuous spectra of fluorophores in biological tissues are the result of the broadening of individual lines. Therefore, the pure spectra are estimated by finding positively defined linear combinations of PCs with the narrowest peaks possible. Still, since components’ spectra significantly intersect, the reconstructed spectra will differ from pure ones. This means that components’ weights will be shifted, but such decomposition correctly describes all experimental data and allows attributing each component to an individual fluorophore.

The detailed analysis of separate excitation–emission matrices showed that, in its turn, each matrix could be described by 3 (maximum 4) principal components (Figure 3a). This means that, in a specific sample, some out of six fluorophores do not excite independently—several fluorophores combine in a single structure in which energy distributes regardless of the excitation wavelength. This is probably one of the reasons why attempts to use tensor PC analysis failed—contributions of fluorophores were not linear.

Since the number of components is significantly reduced, one of them, located in the longer wavelength region, has no other components, defined completely inside its spectral band, and thus could be reconstructed precisely, for example, by using the narrow peak approach. Examples of such fluorescent components’ spectra of similar, measured at two different spots of a single sample, excitation–emission matrices are shown in Figure 3b. These two fluorescent components are different, meaning that they consist of several fluorophores, excited simultaneously in equal proportions in a specific sample regardless of the excitation radiation wavelength. Comparing these two components, it is possible to derive the differential spectrum (Figure 3c), which describes the observed variation. Analyzing in such a manner all the available excitation–emission matrices, it was found that out of six overall components, three separate fluorophores with 400, 440, and 550 nm peaks exist. Their spectra were reconstructed using several matrices. One of the features of the components found is that several of them showed distinct features of fluorescence reabsorption around 420 nm (Figure 3d), which is usually attributed to the oxyhemoglobin presence [21]. The spectra of the revealed longer wavelength components could be clarified using 300–330 nm excitation in the same manner (Figure 4). Figure 5 shows the final reconstructed spectra with 400, 440 and 550 nm peaks after comparing and averaging.

One of the other three components has a peak around 380 nm and cannot be reconstructed precisely since its spectral band completely overlaps spectral bands of the above-mentioned fluorophores. It was reconstructed using the narrow peak approach with the 310 nm excited fluorescence spectra. The same was done with the other two components, which form the main peak. These two spectra were estimated using the narrow peak approach from matrices where the first two principal components are most dominant (Figure 5).

Figure 6a shows an example of the components’ excitation spectra in one of the excitation–emission matrices—dependence of the components’ contributions to overall spectra at the specified wavelengths in the 210–290 nm range. Excitation spectra may significantly vary from sample to sample when tissues are not optically thin; thus, fluorophore contributions are mutually dependent. Non-fluorescent chromophores also can introduce discrepancy. Despite such uncertainty, contributions from 370, 400, 440, and 550 nm peaks correlate; normalized excitation spectra of these components either match in all excitation radiation wavelength ranges or differ only in 260–290 nm ranges, as the example shows in Figure 6b. As it was mentioned above, this is possible for all these fluorophores are not excited independently but form a structure in which absorbed energy is distributed between fluorophores. Studies performed at longer excitation wavelengths [15,16,22,23] show that various molecular groups in collagen could fluoresce in the observed bands. Spectral features are species-specific and strongly depend on the type of tissue from which collagen was taken. This suggests that all these four fluorophores are present in collagen and form a single fluorescent structure. For wavelengths longer than 260 nm, fluorophores start to be excited by themselves and, as a result, show a difference. Relative contributions of these components may carry information about collagen structure. Comparison with peaks of amino acids (315, 330 nm) may show the proportion of elastic components in the tissue. The detected pattern suggests that two excitation wavelengths could be used to perform a correlation analysis of spectral and mechanical data: “structural” 250 nm and “independent” 290 nm.

Figure 7 shows the distributions of the component peaks for different wavelengths. It can be seen from the presented data that, with rare exceptions, these quantities are not normally distributed. Given the fact that the data represent an average sample size (30< and <100), the use of parametric tests, such as the *t*-test, is incorrect in this case. Therefore, we used the nonparametric Mann–Whitney U test to determine statistically significant differences in peak values for cerebral aneurysms of different statuses. Row by row, Table 1 and Table 2 show the results normalized to the value of the peak indicated in the row since, in this way, we obtain information on the contribution of protein or elastic components. In particular, it can be seen that it is precisely short-wave excitation (less than 300 nm) that allows one to obtain the necessary information. Also, using regression analysis, it was shown that the linear regression law (for example, for the 315 nm component at radiation at a wavelength of 250 nm) describes the relationship between ultimate stresses, deformation and component contribution by 48% better than comparing the mean values (Figure 8). Tables with 250-nm and 290-nm wavelength radiation values normalized by the 330-nm component are given in the Appendix A.

## 4. Discussion

In the last decade, the number of studies on the tissue of cerebral aneurysms in living patients has increased significantly. This is primarily due to significant differences between the strength properties of living and cadaveric tissues [22]. Such data are successfully used in preoperative modeling [23,24]. In our work, we have demonstrated a new approach to the non-invasive determination of aneurysm status and its strength properties by means of regression analysis. Given the future technical possibility of non-invasive (minimally invasive) irradiation of aneurysm tissue, most of which are localized in the vessels located on the surface of the brain parenchyma, it would be possible to use laser-induced fluorescence as a method to obtain important data on the strength characteristics of the aneurysm wall material to indicate similar data in calculation packages (ANSYS, Comsol, etc.) and further preoperative modeling. However, despite a number of new results that were obtained for the first time for cerebral aneurysms, the work is not devoid of a number of disadvantages. In particular, at the moment, regression analysis only allows us to determine to which cohort the presented sample belongs: to the cohort of ruptured or unruptured aneurysms. However, from a practical point of view, understanding the rate of migration from one cohort to another in order to perform elective surgery or patient follow-up is of much greater practical interest. It is assumed that this drawback can be eliminated if, in parallel with the accumulation of data on the strength properties of the CA walls and their LIF data, the issue of changing the morphology of the aneurysm (for an unruptured aneurysm) or the time interval from the moment of measuring the LIF and the SAH moment (for ruptured aneurysms) is considered.

Another important point, the implementation of which would significantly help in achieving precisely the quantitative results of the relationship between LIF and strength characteristics, is the study of tissues of healthy intracranial and extracranial arteries of healthy patients. The possibility of access to such samples is currently limited for our team. Such study is associated with a whole range of problems since an investigation can be performed only when, during a destructive vascular surgery on the brain, the length of the donor artery allows cutting off a piece of sufficient size for research. The work [25] indicates the realism of the above-described prospects for using the technique, in which this technology, in fact, could already bring a specific result.

In the work, experiments were carried out with freshly extracted samples of human cerebral aneurysms. Currently, such systems are not used to determine the status of aneurysms; however, in our work, the possibility of such separation was demonstrated for the first time. As already shown in [26], similar technology is already being used in medical practice for the diagnosis of cancer.

## 5. Conclusions

Laser-induced fluorescence excitation–emission matrices of 20 ruptured and unruptured cerebral aneurism samples were measured. Principal component analysis showed that six fluorophores contribute to each spectrum. Using the narrow peak approach and pairwise comparison, spectra of six fluorescent components with peak wavelengths 315, 330, 370, 400, 440, and 550 nm were reconstructed. The excitation efficiencies of the last four correlate for various laser radiation wavelengths, suggesting that all four form a single fluorescent formation, specific to the collagen structure of each sample. The two most informative excitation wavelengths were selected—250 and 290 nm. According to LIF, there are statistically significant differences (*p* = 0.000119) between the material of ruptured and unruptured cerebral aneurysms. The performed regression analysis indicates a close relationship between components at 315 nm and the strength characteristics of the samples studied.

## Figures and Tables

**Figure 1 biomedicines-09-00537-f001:**
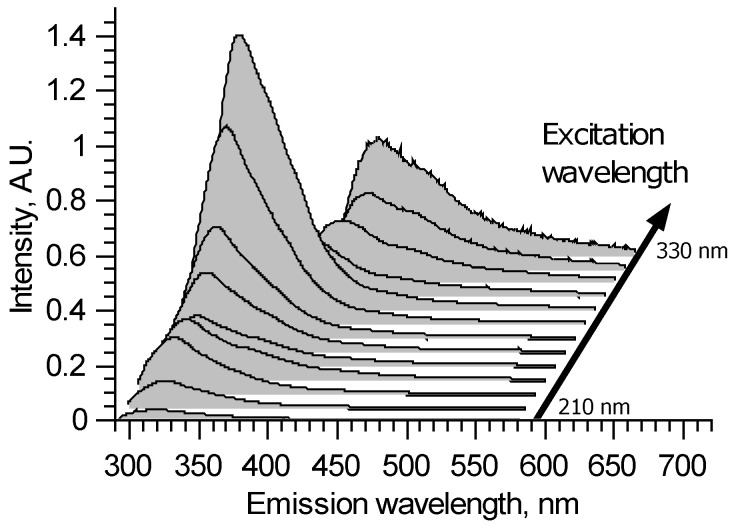
An example of laser-induced fluorescence excitation–emission matrix for artery sample.

**Figure 2 biomedicines-09-00537-f002:**
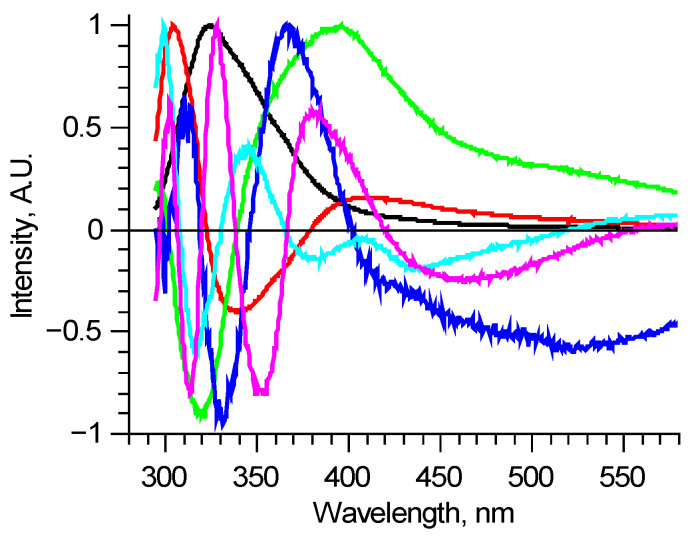
Principal components of all measured laser-induced fluorescence spectra (shown in Figure 1. as an example). Each principal component is an abstract function, and doesn’t directly correspond to real fluorophore.

**Figure 3 biomedicines-09-00537-f003:**
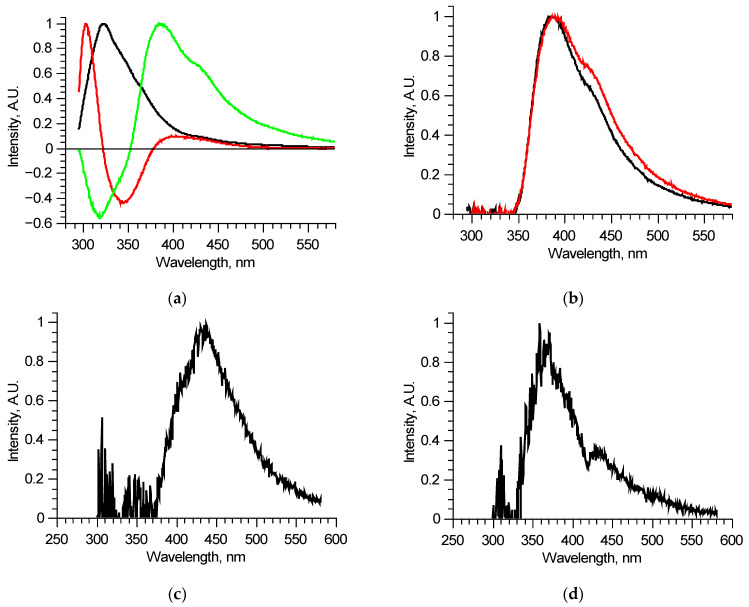
Analysis of separate excitation–emission matrices for 210–290 nm laser radiation wavelengths: principal components of an individual matrix (**a**); reconstructed long wavelength fluorescent component spectra for two similar matrices (**b**); differential spectrum derived from two components (**c**); an example of a reconstructed long wavelength fluorescent component spectrum with signs of reabsorption (**d**).

**Figure 4 biomedicines-09-00537-f004:**
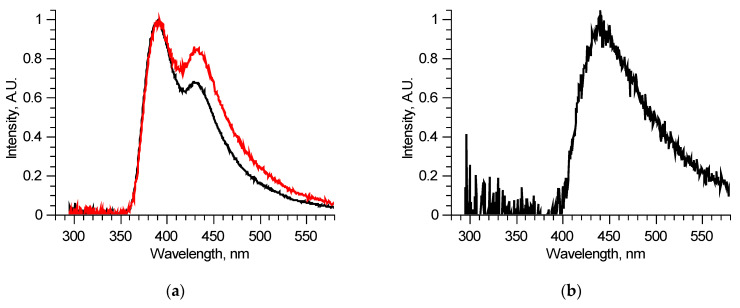
Analysis of separate excitation–emission matrix for 300–330 nm laser radiation wavelengths: spectra of two similar samples (**a**); differential spectrum derived from previous two spectra (**b**).

**Figure 5 biomedicines-09-00537-f005:**
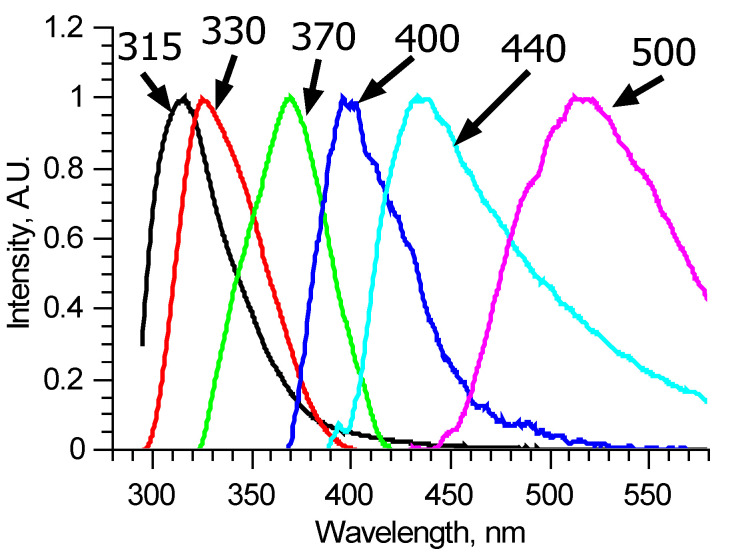
Reconstructed fluorescence spectra of pure components named by their peak wavelengths.

**Figure 6 biomedicines-09-00537-f006:**
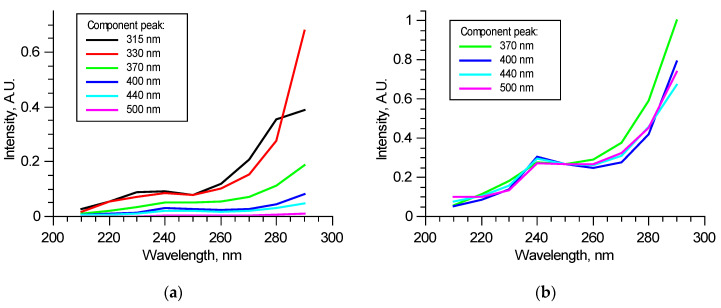
Example of excitation spectra of the specified components in a normalized excitation–emission matrix (**a**) and normalized on 250 nm excitation contributions (**b**).

**Figure 7 biomedicines-09-00537-f007:**
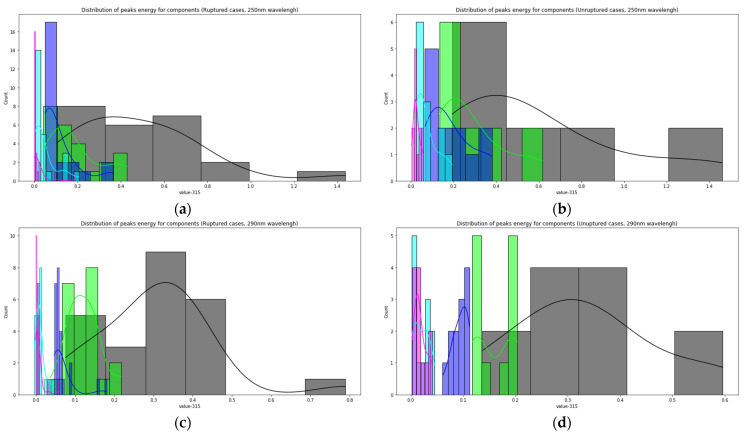
The relationship between the magnitude of the peaks of the LIF components and the status of aneurysms: distribution of peaks energy for components (Ruptured cases, 250 nm wavelength) (**a**); distribution of peaks energy for components (unruptured cases, 250 nm wavelength) (**b**); distribution of peaks energy for components (ruptured cases, 290 nm wavelength) (**c**); distribution of peaks energy for components (unruptured cases, 290 nm wavelength) (**d**).

**Figure 8 biomedicines-09-00537-f008:**
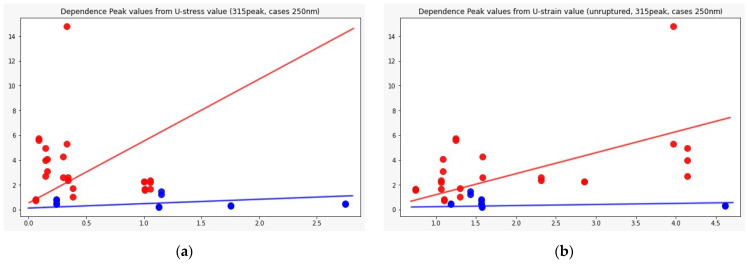
Dependence of the 315 nm peak component value from: ultimate stress (**a**) and ultimate strain (**b**); Red dots: ruptured cases, blue: unruptured. Regression analysis was performed, which showed the best result for the 250 radiation wavelength, 315 peak component, where linear regression, in this case, is 48% more preferable than using mean values (*p*-value = 0.01).

**Table 1 biomedicines-09-00537-t001:** The magnitude of the error of the second kind (*p*-value) for the Mann–Whitney U test for samples of two cohorts: ruptured and non-ruptured aneurysms at a radiation wavelength of 250 nm.

Peak Wavelength/ Normalized by	315	370	400	440	500
315	-	0.055468	0.0042	0.01397	0.000329
330	0.2564	0.0107	0.00228	0.0128	0.000534
370	0.05546	-	0.067676	0.055468	0.0062

**Table 2 biomedicines-09-00537-t002:** The magnitude of the error of the second kind (*p*-value) for the Mann–Whitney U test for samples of two cohorts: ruptured and non-ruptured aneurysms at a radiation wavelength of 290 nm.

Peak Wavelength/ Normalized by	315	370	400	440	500
315	-	0.1869	0.028978	0.0769	0.0002
330	0.34977	0.01175	0.01176	0.026829	0.000119
370	0.18692	-	0.20549	0.05546	0.00119

## Data Availability

The raw data of the LIF and tensile tests are available at the link (we will upload the data on Mendeley).

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
