# Peer review of "The Relationship between the Strength Characteristics of Cerebral Aneurysm Walls with Their Status and Laser-Induced Fluorescence Data"

_biomedicines, 2021, doi:10.3390/biomedicines9050537_

Round 1

Reviewer 1 Report

The relationship between the strength characteristics of cerebral aneurysm walls with their status and laser-induced fluorescence data

Elena Tsibulskaya, Anna Lipovka, Alexandr Chupakhin, Daniil Parshin, and Nikolay Maslov

Submitted to Biomedicines

This manuscript makes an important scientific contribution and should be published after minor revision. The manuscript contains a few cases where the authors’ meaning is not clear; the suggestions below address these and also finetune the wording of the manuscript and are almost all trivial grammatical changes. This paper will be read by an international audience and needs to receive the acclaim that it is due, which can mostly readily be achieved by maximizing its readability for an international scientific audience.

Lines 2- 3: Omit all but first two “the” so that now reads as is given above.

Line 11: Change to “… many works. However, an …”

Line 13: Change to “… 14 patients and 36 samples …”

Line 16: Omit “the” so now reads “… peaks of two components …”

Line 19: Omit “the” so now reads “… basis for non-invasive …”

Line 28: Change “of” to “for” so now reads “… decision for aneurysms …”

Line 30: Omit “the” so now reads “… in modern …”

Line 32: Insert “the” so now reads “… due to the complex layer …”

Lines 35-36: Change to “Its results should allow performing surgery in …”

Line 37: Change “sing” to “using” so now reads “… determined using different …”

Line 41: Insert comma so now reads “… fluorophores, like …”

Line 43: Change to “… radiation below the 300-nm region …”

Line 46: Change to “… using tunable 210-350 nm laser radiation. Two …” In lines 55 and 63, you give the range as 210-350 nm rather than as 210-290 nm.

Line 48: Change to “… using the narrow peak approach to principal component analysis.”

Line 55: Omit “the” so now reads “… range of 210-350 nm.”

Line 57: Change “to” to “toward” so now reads “… pointed toward the tissue …”

Line 57: Change “latter” so now reads “… The stainless steel cell shows almost …”

Line 58: Change “to” to “onto” so now reads “…mirror onto the …”

Line 61: Change “in” to “at” so now reads “… was at the focus …”

Line 62: Omit “The” so now reads “… sample. Excitation-emission matrices …”

Line 64: Insert comma so now reads “… damage, an optical …”

Line 65: Insert comma so now reads “… range, a custom …”

Line 66: Insert comma so now reads “… range, an acetone …”

Line 67: Insert two commas and a hyphen so now reads “… used, providing 330-nm cutoff wavelength. For 320-330 nm range, a high-pass …”

Line 68: Insert comma and hyphen so now reads “…used, with 400-nm cutoff …”

Line 71: Insert comma so now reads “… device, measured …”

Line 72: Change “cm2” to “cm2

Line 73: Change comma to period so now reads “…energy [6]. If this …”

Line 76: Change “could” to “can”

Line 78: Change “in” to “at the” so now reads “… clipping at the Federal …”

Line 81: Insert “a” so now reads “… with a biobath …”

Line 82: Insert “the” so now reads “…resemble the natural …”

Lines 83 and 86: There must be an anion present with the cation Na+.  Is the “sodium solution” a “sodium chloride solution”?

Line 83: Change “37C” to “37 °C”

Line 87: Insert “the”, comma, period, and a capital letter so now reads “… to the laboratory, the geometry … is measured. Then it is …”

Line 88: Omit “the” so now reads “… in specially …”

Lines 90 and 91: Insert space so now reads “1 mm/min” and “2 mm/min”

Lines 93 and 94: Change to “During the first five stages, preconditioning [9] was used, which allows correctly measuring the stresses …”

 Line 97: Change “20” to “Twenty” so now reads “… mm. Twenty samples …”

Line 99: Insert comma and remove second “the” so now reads “… measurements, the specimen … by variability …”

Line 101:  Insert “the” so now reads “… using the following …”

Line 104: Insert “A” so now reads “A typical …”

Line 105: Change “a” to “the” so now reads “… in the form …”

Line 109: Insert “a” so now reads “… show a main peak …”

Line 110: Change “could” to “can” and insert “the” so now reads “… maximum can shift in the 320-330 nm …”

Line 114: Insert “the” so now reads “… In the longer …”

Line 115: Insert comma so now reads “… region, all …”

Line 117: Insert “the” and a comma so now reads “...In the longer wavelength region, the …”

Line 118: Change “It” and insert “to be” so now reads “The 374-380 nm peak appears to be due to …”

Line 121: Insert comma so now reads “… wavelengths, its …”

Line 121: Change “form” to “from”

Line 124: Change so now reads “Thus, collagen fluorescence couldn’t …”

Line 130:  Change to “… allow performing any …”

Line 133: Insert “an” so now reads “… components an extremely …”

Line 135: Change “a priory” to “a priori

Line 139: Insert comma so now reads “… range, Fig. 2 …”

Line 146: Omit “vessel” so now reads “An example of laser-induced fluorescence excitation-emission matrix for _______ sample.” (fill in type of sample in blank)

Line 148: Omit “vessel” so now reads “Principal components of laser-induced spectra shown in Fig. 1.”

Line 154: Change “In” to “For” and change “it” to “this” so now reads “For an optically thin sample, this means that …”

Line 158: Change to “… allow finding positively …”

Line 159: Insert space so change “… components[17].” to “… components [17].”

Line 163: Change to “… allows finding unique …”

Line 169: Change to “… allows attributing each …”

Line 171: Change “Figure 5.” to “Figure 3.”

Line 176: Change to “… by 3 (maximum 4) principal …”

Line 177: Change to “… in a specific sample, some of the six fluorophores …”

Lines 178-179: Insert “of” so now reads “… regardless of the …”

Line 185: Change to “… precisely, for example, by using the narrow peak …”

Line 187: Change “the” to “a” so now reads “… of a single …”

Line 189: Insert “a” so now reads “… in a specific …”

Line 192: Inset comma so now reads “… matrices, it …”

Line 195: Change to “… components found is that several …”

Line 197: Change “found” to “revealed” so now reads “… spectra of the revealed longer …”

Line 205: Change to “One of the other three components …”

Line 206: Change “includes” to “overlaps”

Line 207: Change “to” to “with the” so now reads “… approach with the 310-nm …”

Line 208: Insert “was done” and change “rest” to “other” so now reads “The same was done with the other two …”

Line 210: Insert “the” so now reads “… where the first two …”

Line 211: Insert “the” so now reads “… example of the components’ excitation …”

Line 214: Change dash to “when” so now reads “… sample when tissues are …”

Line 216: Change “of” to “from” and change dash to semicolon so now reads “…contributions from 370, … peaks correlate; normalized …”

Line 218: Change to “…ranges … as shown in the example (Fig. 6b).”

Line 219: Insert comma and change “of” to “for” so now reads “… above, this is possible for all …”

Line 221: Insert “at” and omit comma so now reads “…performed at longer … [14,15,21, 22] show that …”

Line 223: Insert “the” so now reads “…on the type …”

Lines 223-224: Change to “This suggests that …”

Line 225: Insert comma so now reads “…260 nm, fluorophores …”

Line 226: Change to “… themselves and as a result show differences.”

Line 235: Insert two commas so now reads “… tests, such as the t-test, is …”

Line 237: Omit “the” so now reads “Row by row, Tables 1 and 2 …”

Lines 243, 245, 247, 248, 249, and in titles above plots in Fig. 8: Change “nM” to “nm”. The units are nanometers (nm), not nanomolar (nM).

Figure 8: what are the units of the x-coordinates and of the y-coordinates? Either label the axes of the plots or specify the units in the Fig. 8 figure caption.

Line 250: Insert “nm” so now reads “… the 315-nm peak …”

Line 251: Insert “nm” so now reads “… for the 250-nm radiation …”

Line 252: Insert “nm” so now reads “…315-nm peak component…”

Lines 255-256: Change to “… significant differences between the strength … tissues [23].”

Line 266: Insert “only” and remove “rather” so now reads “… analysis only allows us to determine to which …”

Lines 278-280: Change to “… patients. The possibility … for our team. Such a study is … problems, since an investigation can be performed only during …”

Line 287: Insert “the” so now reads “Using the narrow peak approach…”

Line 295: Change to “… component at 315 nm and …”

Author Response

We thank you for the attention to our manuscript. All of the listed suggestions were taken in the account.  

Reviewer 2 Report

This manuscript reports an experimental study of the artery walls with cerebral aneurysms using tunable laser system.The subject is interesting, and the paper is well organized. The author showed the lmechanical and spectroscopic measurements in 14 pa-tients and 36 their samples of CA tissue。Different from the other similar literature reports, The innovation of this paper is to determine the intensity characteristics and status of Cerebral aneurysms through non-invasive means of tunable laser system, At last, a linear regression model has been built that describes the correlation of the magnitude of the ultimate strain and stress with the magnitude of the peaks of one of the components. These findings are of importance and novelty in medical diagnosis fields and I recommend publication in biomedicines Today after minor revision. 1. The power parameters of the adjustable laser system and fluorescently labeled samples should be described in detail. 2. The impact of tunable laser on human safety and its practical application value has not been demonstrated in the article and needs to be further supplemented. 3. The experimental data in the paper is limited and representative. Whether the linear regression model used can be extended to other data needs supplementary explanation on the impact.

Author Response

We thank you for your attention to our manuscript.
Here we would like to reply to the comments:
1. The laser power parameters are described in Materials and Methods section. We have extended the related part of this section adding more information about irradiation properties and its influence on the samples. It is also important, that no invasive fluorescent labeling was used in our experiments, only intrinsic fluorescence. We have added a comment about that in the Introduction section.
2. In the work, experiments were carried out with freshly extracted samples of human cerebral aneurysms. Currently, such systems are not used to determine the status of aneurysms, however, in our work, the possibility of such separation was demonstrated for the first time. As already shown in [http://doi.wiley.com/10.1002/lsm.10191], a similar technology is already being used in medical practice for the diagnosis of cancer.  
We have noted this fact in the end of the Discussion section.
3. In our work, the main task was to find and show differences in LIF results for aneurysms of different status. On the one hand, the differences themselves should be expected, since the difference in the strength properties of aneurysms of different status has already been shown, as well as in their composition according to the data of histological examination. However, our work is the first in which a non-tactile verification method is applied, which in the future, we believe, can be used to determine the status of the aneurysm, or be a method by which it will be possible to construct a map of the composition/strength of the aneurysm tissue, which could be useful in the future development of MR-spectroscopy [https://doi.org/10.3389/fpsyt.2020.00769] and determination of the aneurysm status according to such a study. 
The paper presents data that are the most interesting from a statistical point of view and that are most indicative of the differences in LIF data for aneurysms of different status. All raw data is attached to Mendeley and could be analyzed by other researchers on their own to find possibly more interesting relationships.